

# Technical note: An open-source, low-cost system for continuous nitrate monitoring in soil and open water

Sahiti Bulusu[1], Cristina Prieto García[2], Helen E. Dahlke[2], Elad Levintal[3,*]

[1] Basis Independent Fremont Upper School, Fremont, CA 94539, USA

[2] Department of Land, Air and Water Resources, University of California, Davis, CA 95616, USA

[3] Zuckerberg Institute for Water Research, Jacob Blaustein Institutes for Desert Research, Ben-Gurion University of the Negev, Sde Boker campus, 84990, Israel

*Correspondence to*: Elad Levintal (levintal@bgu.ac.il)

**Abstract.** Nitrate ($NO_3^-$), mainly leaching with soil pore water, is the primary nonpoint source pollutant of
groundwater worldwide. Obtaining real-time information on nitrate levels in soils would allow gaining a better understanding of the sources and transport dynamics of nitrate through the unsaturated zone. However, conventional nitrate detection techniques (e.g. soil sample analysis) necessitate costly, laboratory-grade equipment for analysis, along with human resources, resulting in a laborious and time-intensive procedure. These drawbacks raise the need to develop cost-effective and automated systems for
in situ nitrate measurements in field conditions. This study presents the development of a low-cost, portable, automated system for field measurements of nitrate in soil pore water and open water bodies. The system is based on the spectrophotometric determination of nitrate using a single reagent. The system design and processing software are openly accessible, including a building guide, to allow duplicating or changing the system according to user-specific needs. Three field tests, conducted over five weeks, validated the system's
measurement capabilities within the range of 0-10 ppm $NO_3^-$-N with a low RMSE of <0.2 ppm $NO_3^-$-N when comparing the results to standard laboratory nitrate analysis. Data derived from such a system allow tracking of the temporal variation in soil nitrate, thus opening new possibilities for diverse soil and nutrient management studies.

## 1. Introduction

Nitrogen (N) is a macro-nutrient found in soil, groundwater, and open water bodies across the globe. Nitrogen is essential for crop production and applying nitrogen-based fertilizers is a common practice in agriculture. However, excess fertilization leads to low nitrogen use efficiency (NUE) and can cause groundwater contamination due to leaching of excess nitrate ($NO_3^-$) in the soil, which is the mobile form of nitrogen and is easily transported by water (Ascott et al., 2017; Turkeltaub et al., 2021; Levintal et al.,
2023). Nitrate leaching from agricultural soils through the vadose zone has become the primary nonpoint source pollutant of groundwater (Ascott et al., 2017; Richa et al., 2022; Gurdak and Qi, 2012). Elevated



nitrate concentrations in open water, in addition to groundwater, are also considered a major global threat that can cause algae blooms and loss of aquatic life (Van Metre et al., 2016; Wherry et al., 2021).

Optimizing fertilization by applying the needed amount of nitrogen fertilizer for the crop at each growing stage can reduce the environmental risks above. To achieve this, real-time information on soil pore water nitrate levels is required (Yeshno et al., 2019), leading to a need for an accessible method to measure real-time nitrate concentrations in soils. However, measuring continuous in situ soil pore water nitrate concentrations is still a major environmental and agricultural challenge. During the last two decades, different soil pore water nitrate characterization technologies were tested, including ion-selective electrodes, portable spectrophotometers coupled with suction cups, and lab-on-a-chip technologies (Bristow et al., 2022).

The majority of published nitrate sensing systems for soil pore water show promising directions, however, they are limited to only lab tests, require complicated and repeated calibration procedures, or may be considered as a proof-of-concept rather than a functional field system (e.g., Ali et al., 2019; Chen et al., 2023; Tuli et al., 2009). Only two published studies, as far as we know, showed significant progress in measuring soil pore water nitrate concentrations continuously in the field. Bristow et al. (2022) developed ion-selective electrodes for soil nitrate sensing. The electrodes were field tested under a relatively high nitrate measurement range of ~50-300 ppm $NO_3^-$-N with a reported Root Mean Square Error (RMSE) of ~16 ppm $NO_3^-$-N. They described significant drift after eight weeks of field deployment that required the development of a correction algorithm. In general, electrode fouling, drift, ion interference, limited sensitivity, and the need for temperature compensation are major disadvantages of ion-selective electrodes (Tuli et al., 2009).

Yeshno et al. (2019) presented a monitoring system for continuous measurements of nitrate concentrations in soil pore water. Their system is based on ultraviolet (UV) absorbance spectroscopy to directly determine nitrate without pretreatment of the sample, such as filtration or adding reagents. The system was tested at four agricultural field sites during four sampling campaigns. The nitrate measurement range was ~10-350 ppm $NO_3^-$-N (no RMSE was reported). The main advantage of the system is the durability and lack of needed field calibration, thus we consider this system as the most advanced and robust solution for field nitrate measurements currently available. Yet, the methodology is patented with no assembly details provided, and therefore, it cannot be duplicated and deployed by other users.

This study describes the construction and performance of a portable, low-cost, automated system for pore water nitrate measurements. The system is based on a spectrophotometer coupled with an array of pumps and a suction cup installed in the soil. A comprehensive technical documentation encompassing system



design, assembly, programming, deployment, power management, and data analysis is included to allow
end-users to replicate, modify, and deploy the system to their specific requirements without requiring prior
engineering expertise. For validation, three field tests were conducted over five weeks.

## 2.   Materials and Methods

The system is based on the spectrophotometric determination of nitrate using a single reagent (Doane and
Horwáth, 2003). Each water sample is mixed with a reagent (Vanadium(III) chloride (VCl$_3$) + $N$-(1-
naphthyl)ethylenediamine dihydrochloride (NEDD)) and then measured at 540 nm wavelength. The
absorption intensity is used to determine the nitrate concentration using a calibration curve as detailed
below.

### 2.1.    Hardware

The field nitrate sensing system is established on the open-source hardware concept (Pearce, 2012, 2014)
and consists of three segments: the spectrophotometer, the hydraulics system, and the control unit (**Fig. 1a**).
The low-cost spectrophotometer is based on the design by Laganovska et al. (2020), utilizing the
C12880MA mini-spectrometer chip (Hamamatsu Photonics K.K., Japan). The device measures absorption
in the 450-750 nm range, yet we use only the 540 nm wavelength. The 3D-printed measurement box holds
the cuvette for sample measurements (**Fig. 1b**).

The hydraulics system consisted of six peristaltic dosing pumps, a set of 1/16" (1.57 mm) inner diameter
tubing, a 50 mL container for collecting the initial water sample (#1), a 15 mL container for mixing the
sample with the reagent (#2), a 50 mL container for post-processing waste collection of the sample mixed
with the reagent (#3), a reagent box (#4), and a ceramic suction cup used to collect the water samples from
the tested soil or water body (#5) (**Fig. 1a** and **1b**). The first pump (P1) is connected to the ceramic suction
cup for sample collection. The rest of the pumps work in coordination to mix the appropriate sample volume
with reagents, deposit it in the cuvette, and then clean the tubes and cuvette once the measurement is taken.
The spectrophotometer and hydraulics system are controlled using an open-source microcontroller
(Arduino Mega, Arduino, Italy) with a micro-SD card for data logging. The system is powered by a 12 V,
7 Ah battery connected to a 10 W solar panel. Hardware details, system assembly instructions, and pumps
sequence are provided on our GitHub page https://github.com/SahitiB/AGNET/tree/main.



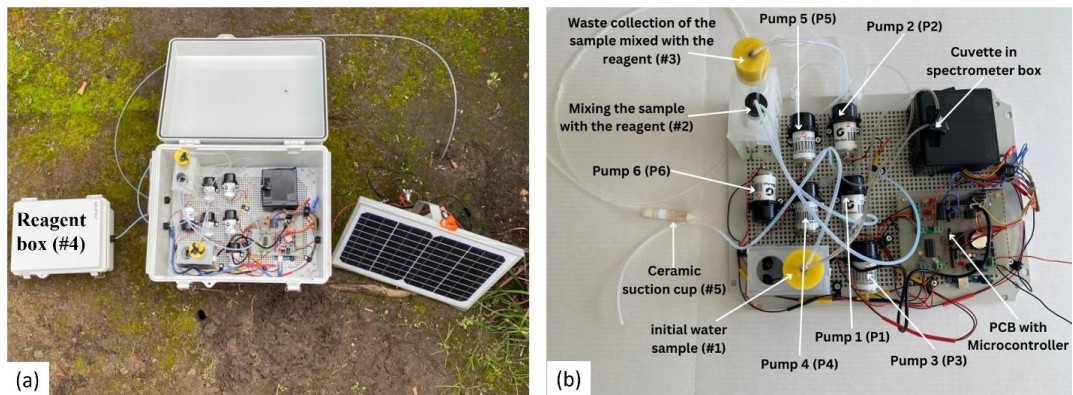

**Figure 1**. Experimental setting. The complete system during soil testing (a) and zoom in on the main box (b). We note that the spectrophotometer unit is based on the design by Laganovska et al. (2020).

## 2.2. Software

An Arduino Mega microcontroller controls the device. Programming the Mega is done using C++, the default language of the Arduino Integrated Development Environment (IDE) (www.arduino.cc/en/software). The code contains the functions required to control the sequence of events for the entire system as well as the process of the spectrophotometer's results. The order and runtime of the pumps are controlled through the code and can be changed as needed. The complete code and open license 100 conditions are described in our Github page https://github.com/SahitiB/AGNET/tree/main.

The flowchart in **Fig. 2** shows the sequence of a single nitrate measurement event, as instructed by the code. First, P1 pumps a soil pore water sample through the ceramic suction cup into container #1 until the water level sensor attached to the container is met at a sample volume of 5 mL. Then, P2 pumps 0.1 mL of the collected sample into container #2, followed by P4 pumping 0.9 mL of the reagent (Vanadium III Chloride) 105 into the same container #2. The mixed sample is then transported to the cuvette (**Fig. 1b**, black box) using pump P3. Once the passive reaction time of 8 hr is completed, the spectrophotometer reading is taken and pumps P5 and P6 vacuum empty out the cuvette and container #1, and the entire system is thoroughly cleaned, emptied, and readied for the next cycle. The user can change the frequency between nitrate measurement events according to needs and battery consumption as detailed below.



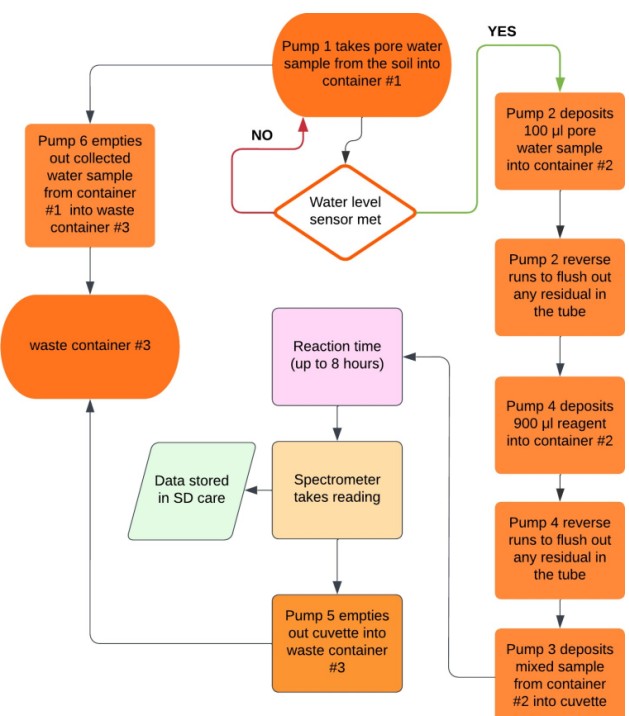


**Figure 2**. A flow chart of the main sequence in a single nitrate measurement event. A more detailed description for each step is given on our GitHub page.

### 2.3. Field deployment

The system was calibrated using standard nitrate samples of 0, 0.1, 0.3, 1, 3, 5, 8, and 10 ppm $NO_3^-$-N. After
running those samples mixed with reagent through the spectrophotometer, a calibration curve (Beer-Lambert curve) is created relating the spectrophotometer absorbance values vs. the standard nitrate samples. A calibration curve is constructed before each experiment. Three field tests were conducted to evaluate system performance – two tests were run on soil pore water samples and one using open water samples. The tests were all carried out in Fremont, California, between February and June 2023. For each test, field-
measured nitrate concentrations were compared against laboratory measurements with a Shimadzu 206-24000-92 UV/Visible scanning spectrophotometer at the University of California, Davis for validation. For the validation, sub-samples were directly taken from the water sample container (#1) after the pore water sample was obtained but before adding the reagent. Comparison of the field-measured nitrate with the standard laboratory method was done in several ways.

First, we conducted the variable test, which aimed at testing the accuracy of the system by randomly varying the amount of nitrate fertilizer in the soil during irrigation (Scotts Liquid Turf Builder with Plus 2 Weed



Control (25% nitrogen content), Scotts, USA). This tested the ability of the system to detect shifts in nitrate levels. Secondly, a continuous test was performed to examine the system stability under rain conditions and the ability to measure nitrate leaching in soil. The variable test ran for seven days with two readings per day, and the continuous test ran for 17 days with one reading per day. In both tests, the suction cup was installed at a depth of 6.3 cm (2.5 in) in the soil, and the system operated autonomously without any maintenance. The third test was the open water pulse test to validate the ability of the system to measure nitrate in water bodies (e.g., rivers and lakes). The suction cup was submerged in a 12 L water bucket for 12 days with one reading per day. Every fourth day, a 0.5 L cup of nitrate-based fertilizer (same as above) was added to the water bucket to test the ability of the system to detect changes in nitrate levels in open water. Atmospheric measurements for the experiments were taken from the California Irrigation Management Information System station (CIMIS; station 171 Union City, CA).

## 3. Results and Discussion

### 3.1. System performances

A summary of the calibration and experimental results is presented in Table 1. One example of the Beer-Lambert calibration curve is displayed in **Fig. 3**. The high $R^2$ of 0.998 between the absorption and the standard nitrate samples validates the linearity of our spectrophotometer and the capability to accurately measure nitrate.

**Table 1**. Summary of the calibration and experiment results

| Experiment type | Duration and sampling rate [d] | Range of tested nitrate [ppm $NO_3^-$-N] | Average RMSE [ppm $NO_3^-$-N] |
|---|---|---|---|
| Calibration | One day | 0-10 | n/a |
| Soil variable test | Seven days (twice per day) 13/2/2023-19/2/2023 | 0-0.97 | 0.09 |
| Soil continuous test | 17 days (once per day) 28/2/2023-16/3/2023 | 0-2.39 | 0.10 |
| Open water pulse test | 12 days (once per day) 10/6/2023-21/6/2023 | 0-7.29 | 0.20 |



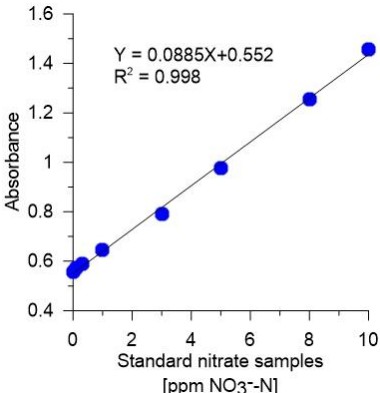

**Figure 3**. Beer-Lambert calibration curve for the system spectrophotometer using known standard nitrate concentrations of 0, 0.1, 0.3, 1, 3, 5, 8, and 10 ppm of $NO_3^-$-N.

The first variable test focused on the low range of nitrate in the soil < 1 ppm $NO_3^-$-N. During the seven-day test, the system was successful in measuring the changes in nitrate level with a RMSE of 0.09 ppm $NO_3^-$-N compared to the lab reference analysis (**Fig. 4a**). We consider this as a low error value that validates our system in the low concentration range of nitrate. The soil continuous test showed similar high accuracy throughout the 17 days of trial with an average RMSE of 0.10 ppm $NO_3^-$-N (**Fig. 4b**). The stability of measurements during this period suggests that no degradation of system performance was occurring with all measurement cycles conducted successfully. This 17-day test was conducted during a significant rain event with 12.2 mm $d^{-1}$ of rainfall occurring on the first day. Therefore, we were able to measure, in real-time, the nitrate leaching in the topsoil (marked by the black curve line in **Fig. 4b**). The third test was conducted to validate the system in open water (**Fig. 4c**). In this case, the suction cup was submerged in a 12 L water container, and nitrate-based fertilizer was added every fourth day. RMSE remained low with 0.20 ppm $NO_3^-$-N. The first step of water sampling using P1 was drastically faster compared to the soil tests, reducing the pump time from 30-40 min to 5-10 min per cycle.





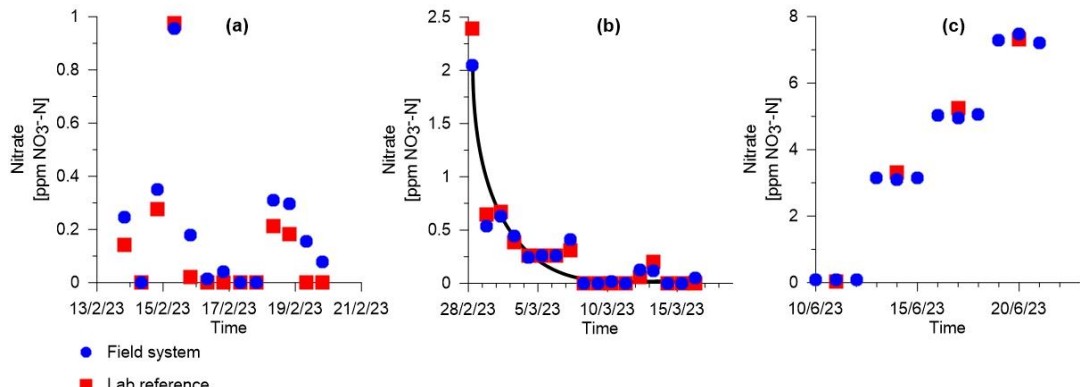

**Figure 4.** Experimental results of the spectrophotometer testing for the soil variable test of low nitrate concentrations (a), the soil continuous test under a rain event (b), and the open water pulse test (c). The black line in subplot (b) marks the nitrate leaching after a rain event of 12.2 mm d$^{-1}$ on 28/2/2023. For each concentration level in the open water pulse test in subplot (c), three field measurements (blue dots) were compared to one laboratory measurement (red squares).

### 3.2. System limitations and modifications

In this study, we presented and tested a portable, low-cost field nitrate sensing system to measure in situ nitrate concentrations in soil pore water and aquatic environments. Although the system is autonomous in terms of obtaining a sample and processing it until a nitrate concentration is determined, it does require some user input. The main user input required involves replacing the dry ice in the reagent box to maintain the recommended temperature of 4 °C for the Vanadium III Chloride reagent to work at its optimum (**Fig. 1a**). In our experiments, replacing the dry ice every five days was sufficient to ensure this temperature, however, all experiments were conducted at an average daily air temperature of 10 °C with a daily maximum of ~20 °C. Warmer conditions will require a more frequent replacement time of the dry ice or adding a cooler box or more advanced solutions, such as a small field refrigerated unit based on solar panel or gas. Improving the reagent chill-box will increase system cost yet reduce human dependency. This, together with the addition of a low-cost modem or wireless communication such as Wifi or LoRa (Levintal et al., 2021; Sanchez-Iborra et al., 2018), will make the device completely autonomous for weeks to months.

The system, considering our sample to reagent ratio, can accurately measure nitrate concentrations up to ~10 ppm NO$_3^-$-N. This is a well-known limitation of using the Vanadium III Chloride reagent, which also exists in the lab. When analyzing samples with higher concentrations, adding the reagent turns the water sample to colors that cannot be accurately measured and calibrated using known concentration standards and the spectrophotometer (Doane and Horwath, 2003). A possible solution could be the addition of a visual





color sensor to notify the user when the color is out of range (i.e., high nitrate concentrations) to then dilute the sample accordingly. This will require changing the design of the system and adding a dilution mechanism.

If a high measurement frequency is required, a heating device can be added, or a temperature curve can be developed to reduce the 8-hour reaction time of the Vanadium III Chloride reagent in the cuvette. Yet, this need is highly site-specific as warmer climates will reduce the sample-reagent time naturally. Higher measurement frequency means higher power consumption, which should be optimized using a larger battery capacity and/or a larger solar panel. In our experiments, taking the soil variable test as a reference (**Fig. 4a**), the 12 V, 7 Ah with a 10 W solar panel was sufficient for two samples per day for seven

continuous days. This was achieved under cloudy skies with an average daily solar radiation of 156 W m$^{-2}$. Power consumption is site-dependent due to the variability in the solar panel's efficiency to charge a 12 V battery, and moreover, due to the changes in soil moisture. Lower soil moisture will increase the run time of the peristaltic pump extracting the water sample from the soil (P1 in **Fig. 1**), therefore increasing power consumption for each nitrate sampling cycle. In very dry conditions, water samples cannot be extracted

from the soil and the system will not work. We note that this is a common problem of using suction cups in dry soils unrelated to this specific system.

This study demonstrates the capabilities to measure nitrate leaching during a rain event and nitrate changes in open water. Additional potential research objectives for the low-cost portable nitrate system include: (1) measuring soil nitrate levels in the root zone of an agricultural field during a growing session to optimize

nitrogen fertilization applications, i.e., precision agriculture methods to reduce groundwater pollution (Yeshno et al., 2019). This application will need to include a soil moisture sensor to allow the calculation of the nitrate stock available for plant uptake (Bristow et al., 2022); (2) couple the system with low-cost oxygen sensors (Levintal et al., 2022) to investigate in real time the occurrence of denitrification and its dependency on soil oxygen levels (Levintal et al., 2023); (3) measure nitrate changes in lakes/rivers during

heavy rain events or floods, and (4) implementing the same design to measure other contaminants in the soil pore water and open water given that they have distinct absorbance in the range of our spectrophotometer of 450-750 nm.

## 4. Summary

This study presents the development of a low-cost, portable, automated system for field measurements of

nitrate in soil pore water and open water bodies. The system consists of an Arduino-controlled array of pumps, a suction cup installed in the soil, and a spectrophotometer that measures the nitrate concentration after the water sample is mixed with a reagent. Three field tests conducted over five weeks to validate the



system within a measurement range of 0-10 ppm $NO_3^-$-N showed a low RMSE of <0.2 ppm $NO_3^-$-N when comparing the results to standard laboratory nitrate analysis. The system design and processing software

are openly accessible. By designing a system in which all electronics are limited to buyable hardware components and the files for the printed circuit board (PCB) are provided, it is possible to duplicate or change the system according to user-specific needs. The total cost of the system components is USD $1,100, excluding reagents, which we hope will allow reproducibility and open new possibilities for conducting field studies in soil and environmental nitrate monitoring.




**Code and data availability**

The complete technical guide and code are available in our GitHub repository (https://github.com/SahitiB/AGNET/tree/main)

**Author contributions**

EL conceptualized the study and designed the system, SB constructed the system and conducted the study, SB and EL wrote the first manuscript draft, HED provided the resources and project supervision. All the authors (SB, CPG, HED, and EL) contributed to the final version.

**Competing interests**

The contact author has declared that none of the authors has any competing interests.

**Acknowledgments**

This research has been supported by the Gordon and Betty Moore Foundation (grant no. 7975), the United States–Israel Binational Agricultural Research and Development Fund (Vaadia-BARD Postdoctoral Fellowship no. FI-605-2020), and the USDA National Institute of Food and Agriculture grant no. 2021-68012-35914.



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
