# Peer review of "Technical note: An open-source, low-cost system for continuous"

_EGUsphere, 2023_

## Author Comment (AC1)

Referee #1

The manuscript presents a technical setup for nitrate measurements in soil and open water in field conditions. Obviously, the subject is super important and critical in many scientific and environmental aspects. On one hand nitrogen pollution that is attributed to excess fertilization in agriculture is one of the main reasons for freshwater disqualification. On the hand the mechanisms related to the dynamic of nitrogen uptake and transport in the soil are also far from being well understood to achieve optimal fertilization. Therefore, any attempt to develop tools for filed measurement of nitrate in soil and water is important and should be published.

In general, the presented methodology involves a sophisticated setup which includes a set of 6 mini peristaltic pumps that synchronize several steps which are managed by an open-source controller. Essentially the main steps include soil porewater sampling into a sampling tank where a reagent (that needs to be refrigerated in the field) is added to the sampled water until equilibrium is achieved (hours). Then the mixed sample is delivered into a cuvette for spectral analysis through synchronized automated process. In spite of its complexity and non-trivial field applicability the authors prove that it is doable and show reasonable results. Nevertheless, I have a major concern regarding the presented context and applicability of the proposed methodology.

The main rational for developing the system is measuring soil nitrate concentration in agricultural setups. However, the measurement concentration range of the method presented here is inherently very low, limited to 10 ppm N-NO3 (which is the max allowed concentration in drinking water). Nevertheless, soil nitrate concentration in fertilized agriculture is mostly far above that concentration, ranging between tens to hundreds ppm. Indeed, throughout their final chapter the author acknowledges this concentration limit. Nevertheless, I think that this limitation should be stated upfront in a clear manner, from the title and abstract to the objectives. Beyond this comment I want to state again that it is an elegant method that merit publication after revision that put in context the limited applicability.

We want to thank the reviewer for the positive feedback. Following the main comment on the measurement range, we revised the text in the following sections:

(1) The title was changed to emphasize the system's low range of nitrate:

"Technical note: An open-source, low-cost system for continuous monitoring of low-concentration nitrate in soil and open water".

(2) We added the range of our system at the end of the introduction:

"For validation, three field tests with a concentration range of 0-10 ppm $NO_3^-$-N were conducted over five weeks."

(3) The discussion on this limitation was extended in section 3.2 (System limitations and modifications):

"The system, considering our sample to reagent ratio, can accurately measure nitrate concentrations up to ~10 ppm $NO_3^-$-N. This is a well-known limitation (Doane and Horwath, 2003) of using the Vanadium III Chloride reagent, which also exists in the lab. It would be possible to increase the range with the current system by either increasing the amount of reagent, decreasing the amount of sample, or both. However, further experiments would be necessary to implement and test this extended range. When analyzing samples with higher concentrations (70-80 ppm $NO_3^-$-N), adding the reagent results in unusual colors (salmon, orange, and yellow) that cannot be accurately measured and calibrated using known concentration standards and the spectrophotometer. For example, a sample with a high concentration (e.g., approximately 120 ppm $NO_3^-$-N) will turn yellow and the spectrophotometer will register absorbance corresponding to concentrations lower than 1 ppm $NO_3^-$-N, indicating a false reading. A possible solution could be the addition of a visual color sensor to notify the user when the color is exceeding the concentration range covered by the standard (i.e., high nitrate concentrations) to then dilute the sample accordingly. This will require changing the design of the system and adding a dilution mechanism."

(4) We added the applicability limitation to the conclusion section:

"This nitrate range is suitable for soils with low nitrate concentrations or open water"

---

## Author Comment (AC2)

Referee #2

A very interesting paper looking at taking laboratory testing techniques out into the field and automating them. There are several drawbacks to their current methods (which they do address in the paper, but perhaps could be expanded upon):

We want to thank the reviewer for the positive feedback. Detailed answers are given below using blue font.

- low range - only 0-10 ppm $NO_3^-$ - this is a very low nitrate level in an agricultural setting where we would expect levels a magnitude or two higher than this. The use of dilution to compensate for this is discussed.

Following this comment on the measurement range that the other reviewer also raised, we revised the text in the following sections:

(1) The title was changed to emphasize the system's low range of nitrate:
    "Technical note: An open-source, low-cost system for continuous monitoring of low-concentration nitrate in soil and open water".

(2) We added the range of our system at the end of the introduction:
    "For validation, three field tests with a concentration range of 0-10 ppm $NO_3^-$-N were conducted over five weeks."

(3) The discussion on this limitation was extended in section 3.2 (System limitations and modifications):
    "The system, considering our sample to reagent ratio, can accurately measure nitrate concentrations up to ~10 ppm $NO_3^-$-N. This is a well-known limitation (Doane and Horwath, 2003) of using the Vanadium III Chloride reagent, which also exists in the lab. It would be possible to increase the range with the current system by either increasing the amount of reagent, decreasing the amount of sample, or both. However, further experiments would be necessary to implement and test this extended range. When analyzing samples with higher concentrations (70-80 ppm $NO_3^-$-N), adding the reagent results in unusual colors (salmon, orange, and yellow) that cannot be accurately measured and calibrated using known concentration standards and the spectrophotometer. For example, a sample with a high concentration (e.g., approximately 120 ppm $NO_3^-$-N) will

turn yellow and the spectrophotometer will register absorbance corresponding to concentrations lower than 1 ppm $NO_3^-$-N, indicating a false reading. A possible solution could be the addition of a visual color sensor to notify the user when the color is exceeding the concentration range covered by the standard (i.e., high nitrate concentrations) to then dilute the sample accordingly. This will require changing the design of the system and adding a dilution mechanism."

(4) We added the applicability limitation to the conclusion section:

"This nitrate range is suitable for soils with low nitrate concentrations or open water"

- use of dry ice for cooling, perhaps a peltier cooling plate could be used (although this would have much higher power requirement)

We added this suggestion as another possible solution:

"Warmer conditions will require a more frequent replacement time of the dry ice or adding a cooler box or more advanced solutions, such as a small field refrigerated unit or a Peltier cooling plate based on solar panel or gas."

- relies on soil-pore water being available - would not work in dry soil conditions. Measuring soil moisture/temperature as well would be beneficial.

We added this suggestion to the modification section:

"In very dry conditions, water samples cannot be extracted from the soil and the system will not work. A possible optimization solution could be the addition of a soil moisture sensor to deactivate the system under very dry conditions. We note that this is a common problem of using suction cups in dry soils unrelated to this specific system."

References: include Bristow et al, 2022 in the discussion on LoRa sensing (section 3.2, 1a), as that work developed a remote LoRaWAN nitrate sensor

Done:

"This, together with the addition of a low-cost modem or wireless communication such as Wifi or LoRa (Bristow et al., 2022; Sanchez-Iborra et al., 2018; Levintal et al., 2021), will make the device completely autonomous for weeks to months."

Overall a good, well written paper which presents a new automated field testing utilising existing laboratory techniques. It might be that this system is better suited to open water testing rather than in an agricultural situation, due to the requirement for water to sample and the low measurement range. It was good to see a github resource.